# SPARSE-COMPLEMENTARY CONVOLUTION FOR EFFICIENT MODEL UTILIZATION ON CNNS

## ABSTRACT

We introduce an efficient way to increase the accuracy of convolution neural networks (CNNs) based on high model utilization without increasing any computational complexity. The proposed sparse-complementary convolution replaces regular convolution with sparse and complementary shapes of kernels, covering the same receptive field. By the nature of deep learning, high model utilization of a CNN can be achieved with more simpler kernels rather than fewer complex kernels. This simple but insightful model reuses of recent network architectures, ResNet and DenseNet, can provide better accuracy for most classification tasks (CIFAR-10/100 and ImageNet) compared to their baseline models. By simply replacing the convolution of a CNN with our sparse-complementary convolution, at the same FLOPs and parameters, we can improve top-1 accuracy on ImageNet by 0.33% and 0.18% for ResNet-101 and ResNet-152, respectively. A similar accuracy improvement could be gained by increasing the number of layers in those networks by $\sim 1.5\times$.

## 1 INTRODUCTION

Object recognition has achieved significant improvement through convolutional neural networks (CNNs), e.g., ResNet (He et al. (2016a)) and DenseNet (Huang et al. (2017)). In order to achieve competitive performance on large-scale datasets such as ImageNet (Russakovsky et al. (2015)), these networks usually require increasing model capacity substantially with a deeper network, e.g. ResNet-200, or a wider network, e.g. DenseNet-161. Nonetheless, it seems pretty obvious that the redundancy in those models are still high (Mao et al. (2017)).

This observation encouraged us to investigate an efficient way to increase model utilization[1] for complex architectures. It has been demonstrated that there are multiple ways to increase model utilization, e.g., by keeping the same performance while reducing resource usage (Wen et al. (2017); Kim et al. (2016); Ioannou et al. (2016)) or increasing the accuracy with the same resource usage.

There are two major approaches to increase mode utilization from a macro-architecture[2] of a network perspective. ResNet adds the residual shortcut to assure that the loss at the end can be propagated to all parameters smoothly; thus, all parameters are treated equally for better utilization. DenseNet reuses the extracted features intensively from different layers to save the computations and parameters while achieving good performance. However, both these approaches still rely on regular convolution in their micro-architectures. We found out that deeper and/or wider macro-architectures naturally introduce more redundancy on convolutional layers; the types of convolution can be optimized for such macro-architectures.

We propose a pair of deterministic sparse-complementary convolutional kernels in either the spatial-wise or channel-wise domain to reduce the complexity of each kernel; each sparse kernel has a complementary kernel to approximate the receptive field of regular kernels. Since a sparse kernel saves the floating-point operations (FLOPs) and parameters for a convolution, we are able to enrich

---

[1]The model utilization is defined as the proportion of model performance over the amount of required resources (computations and parameters). E.g., suppose that networks A and B achieve the same accuracy, but A needs more parameters and computations than B; in this case, B achieves higher model utilization.

[2]*Macro-architecture* denotes the topology of a network; *micro-architecture* means the topology within a layer.

Table 1: Comparison with related works.

| Related works | Key features | Orthogonality to ours[*] |
|---|---|---|
| Han et al. (2015); Liu et al. (2015); Collins & Kohli (2014) | Random sparsity | Y |
| Wen et al. (2016); Zhou et al. (2016); Alvarez & Salzmann (2016); Mao et al. (2017); Liu et al. (2017); Wen et al. (2017) | Random but structural sparsity | Y |
| Denton et al. (2014); Jaderberg et al. (2014); Zhang et al. (2015); Kim et al. (2016) | Low-rank approximated kernels | Y |
| Mamalet & Garcia (2012); Szegedy et al. (2016); Sun et al. (2016) | Deterministic low-rank kernels | N |
| Ioannou et al. (2016); Iandola et al. (2016); Ioannou et al. (2017) | Mixed-shape kernels | Y |
| Ours, sparse-complementary kernels | Mixed-shape kernels and deterministic sparsity | – |

[*]: Orthogonality denotes whether or not their works conflict with ours.

feature representation by adding either layers or kernels; therefore our models achieve better model utilization and accuracy under the same resource budget.

The contributions of this paper include the following:

1. We achieve better model utilization on all models with the ResNet and DenseNet macro-architectures on CIFAR-10/100 and ImageNet. We gain $0.33\%$ and $0.18\%$ for ResNet-101 and ResNet-152 on ImageNet, respectively, without increasing any FLOPs and parameters. Such gain would have required adding $1.5\times$ more layers to those networks. Notice that ResNet-152 only improves $0.44\%$ top-1 accuracy than ResNet-101.

2. We propose a better micro-architecture for CNNs. The proposed sparse-complementary kernels save computations and parameters due to their sparsity. The complementary property of a kernel compensates for receptive field missed from the other sparse kernel.

3. We enrich the feature representation under identical resource budget thanks to the sparse and complementary properties in the kernels.

4. We prove that the proposed sparse-complementary convolution achieves the expected speed on the current NVIDIA GPU.

## 2 RELATED WORK

To increase model utilization, several approaches have been proposed that reduce the model parameters and computations with comparative performance; however, our proposed method consists of improving performance with the same model parameters and computations. Table 1 highlights the distinct features of our work, compared to previous research approaches in this space.

**Random Kernel Sparsity** remove unimportant weights based on a trained model. In general, this approach imposes different weight regularization to increase parameter utilization. Some of the research work in this area enforces the zeros into a certain shape, such as forcing all zeros to locate in the same kernel or the same channel within a kernel (Wen et al. (2016); Zhou et al. (2016); Liu et al. (2015); Alvarez & Salzmann (2016); Wen et al. (2017); Liu et al. (2017)); or scattering zeros everywhere (Han et al. (2015); Mao et al. (2017); Collins & Kohli (2014)). Typically, this approach needs a pre-trained model; re-training is then required to recover performance after pruning weights; however, some of them can directly train from scratch by chaning the regularizer in the objective function. Nonetheless, the random sparsity might provide limited advantages on reduction of model size and computations, e.g. the indexed table for non-zero weights or efficient computations in sparse kernels are challenges in terms of implementation overhead. Even though some of works use random

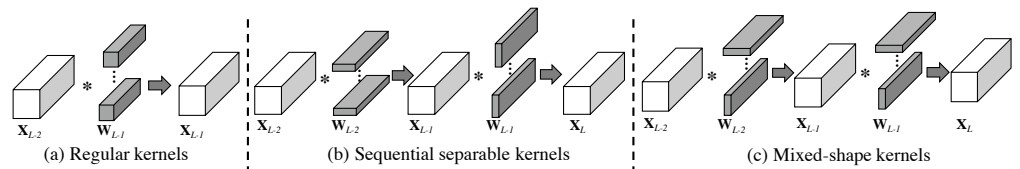

(a) Regular kernels      (b) Sequential separable kernels      (c) Mixed-shape kernels

Figure 1: Different approaches for kernel approximation. (a) Regular convolution, (b) Sequential separate kernels and (c) Mixed-shape kernels. For (b) and (c), the shape of kernels can be arbitrary shape, not limited to $1 \times k$ or $k \times 1$ kernels. The major difference between (b) and (c) is that (c) mixes different shapes of kernels in one layer to extract different features and then the next layer can fuse those distinct features rather than extracting one-type features once, as in (b).

but structural sparsity and gain the speed up over CPU or GPU; however, due to the random structure, it still limits its advantages when deploying on customized neural network accelerator, which might be implemented by ASIC and FPGA. Conversely, thanks to the deterministic and complementary characteristics of our sparse convolution, we can easily achieve competitive speed even with the current CUDNN 6.0. Furthermore, since our method is orthogonal to those approaches, we can integrate them into our solution to increase model utilization.

**Low-rank Approximated Kernels** deploy the factorization on learned kernels into low-rank kernels, or directly deploys hand-crafted low-rank kernels. Tensor factorization can be done by decomposing the kernels in a pre-trained model into multiple low-rank kernels by high-order tensor approximation (Denton et al. (2014); Kim et al. (2016)), singular value decomposition (Liu et al. (2015)), data-driven low-rank approximation (Jaderberg et al. (2014)), or non-linear approximation (Zhang et al. (2015)). By lowering the rank of weight tensors, the redundancy within/between weight tensors can be removed to make the model more compact for higher model utilization. On the other hand, (Mamalet & Garcia (2012); Szegedy et al. (2016); Sun et al. (2016); Fan et al. (2017)) deploy hand-crafted low-rank kernels to approximate high-rank kernels, but the information lost in the low-rank kernels may be unrecoverable. Figure 1 (b) shows an example that sequentially uses low-rank kernels at consecutive layers as full-rank kernels. Our approach uses complementary kernels to compensate the missed information from another sparse kernel pair. Furthermore, since our approach is orthogonal to the factorization approach, our proposed kernels can be factorized into lower-rank kernels to increase model utilization.

**Mixed-shape Kernels** utilize multiple shapes kernels in a convolutional layer (Ioannou et al. (2016); Iandola et al. (2016)) as shown in Fig. 1 (c). Ioannou et al. (2016) uses two 1-D kernels and one optional 2-D kernel, and then combines the extracted features via a $1 \times 1$ convolution. The $1 \times 1$ convolutional layer plays an important role to compensate the disadvantage of using non-complementary kernels. In contrast, our approach replaces their $1 \times 3$ and $3 \times 1$ kernels with sparse-complementary kernels to enrich the feature representations of a network; the complementary characteristic of our networks do not need an additional $1 \times 1$ convolutional layer, which is not optimized on current GPUs.

## 3 PROPOSED SPARSE-COMPLIMENTARY CONVOLUTIONAL KERNEL

The current design of a convolutional kernel in a CNN is inefficient in utilizing parameters of micro-architectures, which results in unnecessary computations and parameters. Several works had demonstrated that, e.g., the model of a CNN can be easily pruned (Han et al. (2015); Mao et al. (2017); Molchanov et al. (2017)) or factorized but keep comparative accuracy (Zhang et al. (2015); Kim et al. (2016)) regardless of its macro-architecture, including, VGGNet, ResNet and DenseNet. Therefore, we became interested in designing CNN micro-architecture that uses the right amount of parameters while sustaining the same accuracy under the same computations. We present a sparse-complementary convolution kernel which can improve accuracy while keeping a similar number of computations and parameters. By simplifying each convolutional kernel with deterministic sparse pattern which uses almost half of parameters of regular kernels, we use saved parameters and FLOPs to add either more convolutional kernels (wider network) or convolutional layers (deeper network) to enrich feature representation; in addition to it, the hardware-friendly deterministic sparsity allows us to achieve almost theoretical speed-up on GPU with CUDNN 6.0. In this paper, two types of sparse kernels are proposed, spatial-wise sparse-complementary (SW-SC) kernel and channel-wise sparse-complementary (CW-SC) kernels, to achieve accuracy improvement under the same resource for the most recent network architectures, ResNet and DenseNet.

In following sections, we explain the sparse-complementary kernels and discuss its advantages with respect to their receptive field coverage; and then, we discuss how we use the saved resource in computations and parameters to improve accuracy compared to its baseline.

### 3.1 SPARSE-COMPLEMENTARY KERNEL DESIGN

We introduce two types of sparse-complementary kernels in this chapter, such as spatial-wise and channel-wise sparse-complementary kernels. The sparse kernel is designed to extract the feature efficiently and its complementary pattern assures the sparse convolution has no missed regions in

receptive field across layers. The two complementary sparse kernels are always paired in one convolutional layer. Let $\mathbf{W}$ represents the weights of a regular $k \times k$ convolutional layer with $C$ channels and $N$ kernels; thus $\mathbf{W}$ is a 4-D tensor, and $\mathbf{W}(x, y, c, n)$ denotes the weight of the $n$-th kernels at the $c$-th channel and spatially located at $(x, y)$ locations, where $x$ and $y$ range from 0 to $k - 1$, where $k$ is kernel size.

The spatial-wise sparse-complementary (SW-SC) convolutional kernel sparsifies the kernel in the spatial domain. Figure 2 (a) and (b) show an example of SW-SC kernels for $3 \times 3$ convolution. Those kernels are either even- and odd-indexed in the spatial domain for all channels in a kernel except for the center point[3]. When kernel size is $3 \times 3$, the kernel shape becomes $+$-shape ($\mathbf{W}_{sw,even}$) or $\times$-shape ($\mathbf{W}_{sw,odd}$). If we formulate the kernel with regular kernel coordinate, the locations with zero weights are (for a convolutional layer with $N$ kernels):

$$\begin{aligned} \mathbf{W}_{sw,even}(x, y, c, 2m) = 0, \text{ if } (y \times k + x) \bmod 2 \neq 1 \text{ and } (y \neq \lfloor k/2 \rfloor \text{ and } x \neq \lfloor k/2 \rfloor), \\ \mathbf{W}_{sw,odd}(x, y, c, 2m + 1) = 0, \text{ if } (y \times k + x) \bmod 2 \neq 0 \end{aligned} \quad (1)$$

where $m < \frac{N}{2}$, and other non-zero locations are trainable weights. Figure 2 (a) and (b) show the kernels in both 3D and 2D views for $3 \times 3$ SW-SC kernels.

On the other hand, with the similar idea, the channel-wise sparse-complementary (CW-SC) convolutional kernel sparsifies the kernel in channel domain, and a pair of kernels are complemented to each other. Thus, one of kernels would take even-indexed feature maps as an input for convolution and the other kernel would use odd-indexed feature maps. The locations of zero weights can be equivalently represented as (for a convolutional layer with $N$ kernels):

$$\begin{aligned} \mathbf{W}_{cw,even}(x, y, c, 2m) = 0, \text{ if } c \bmod 2 \neq 1, \\ \mathbf{W}_{cw,odd}(x, y, c, 2m + 1) = 0, \text{ if } c \bmod 2 \neq 0 \end{aligned} \quad (2)$$

where $m < \frac{N}{2}$, and other non-zero locations are trainable weights. Figure 2 (c) shows an example for $k \times k$ CW-SC kernels. The major difference between CW-SC and group convolution (Krizhevsky et al. (2012) is the order of output feature maps. Thus, if two CW-SC layers are deployed consecutively, it is different from deploying two consecutive group convolutions. E.g., for CW-SC, every feature map at layer L+2 gets the information of all feature maps at layer L; however, for the group convolution, the feature map at layer L+2 only gets the information from either the first half or the second half feature maps at layer L. In short, we can consider CW-SC embeds channel reordering on the output feature maps.

### 3.1.1 Receptive Field of Proposed Kernels

The proposed complementary kernels are designed to cover the identical receptive field that regular kernels can cover when they are applied across several layers. Take stacking multiple $3 \times 3$ convolutional layers as an example, Fig. 3 compares the effective receptive field for different types of

---

[3]The kernels in a pair are spatially complemented to each other on all positions except for the center point.

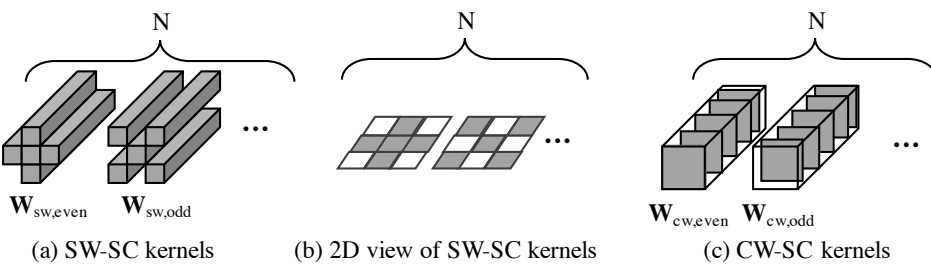

(a) SW-SC kernels      (b) 2D view of SW-SC kernels      (c) CW-SC kernels

Figure 2: Kernel representations of proposed SW-SC and CW-SC kernels ($3 \times 3$ kernel), gray color denotes the locations of trainable weights. (a) 3D-view of SW-SC kernels, and (b) 2D-view of SW-SC kernels, for a $3 \times 3$ convolution, the shapes are $+$ and $\times$. (c) 3D-view of CW-SC kernels, each kernel either samples even- or odd-indexed feature maps for convolution.

kernels and different orders of deployment (Fig. 1 (b) and (c)). We discuss the effective receptive field of a location (across multiple channels) at Layer $L$ from two previous layers (Layer $L-1$ and $L-2$). Figure. 3 (a) is the regular one, stacking conventional layers, and its effective receptive field is the intact one, i.e. covering everything as expected.

Nonetheless, if we *sequentially* apply sparse kernels (even-indexed ($\mathbf{W}_{sw,even}$) and odd-indexed ($\mathbf{W}_{sw,odd}$)), like in Fig. 1 (b) (Fan et al. (2017)), the effective receptive field is narrow than the conventional one. Figure 3 (b) and (c) show the results for applying $\mathbf{W}_{sw,even}$ and $\mathbf{W}_{sw,odd}$ in different orders. Moreover, if we do not apply sparse and complementary kernel alternatively, more receptive fields are uncovered.

For the case of SW-SC or CW-SC kernels, we can achieve the same receptive field since the convolution operates across input channels. That is, the feature maps at layer $L-1$ is derived from both odd-indexed ($\times$-shape for $3 \times 3$ kernel) kernels and even-indexed ($+$-shape for $3 \times 3$ kernel) kernels at layer $L-2$. Then, when applying one of sparse and complementary kernels on feature maps of the $L-1$-th layer, the produced features are able to contain whole receptive field from layer $L-2$. Figure 3 (d) and (e) (corresponding to (b) and (c)) show the receptive field of applying SW-SC kernels. Thanks to the complementary characteristic, most uncovered receptive fields are recovered (the yellow blocks in Fig. 3.) Therefore, it again justifies the advantages of complementary feature in designing kernels. The effective receptive field of CW-SC can be derived in a similar approach. The experimental results show that utilizing complementary patterns within the same layer is better than applying single pattern within one layer (see Section 4.1.4).

## 3.2 Enriching Feature Representation under Identical Computational Budget

Enforcing deterministic sparsity in either spatial or channel domain enables us to utilize the saved resource budget in both computations and parameters for enriching feature representations in advance. The straightforward idea is either to increase the number of layers or the number of kernels in a layer. He & Sun (2015) compared the performance of wider or deeper network; however, their conclusions show that deeper network is better than the wider network for most cases, but at certain depth level, the deeper network performed worse than the shallow network. This is because they did not test on the ResNet architectures, the deeper network might have suffered from gradient vanishing problem. On the other hand, WideResNet (Zagoruyko & Komodakis (2016)) discussed the advantages of increasing the number of kernels in a convolutional layer with respect to accuracy and computational efficiency; furthermore, current NVIDIA GPU is more computational-friendly for parallelism from multiple kernels within a layer rather than multiple layers with a small number of kernels. Under similar computational load, the wider networks are always more efficient than deeper networks for NVIDIA GPU. Details are discussed in Section 4.1.3.

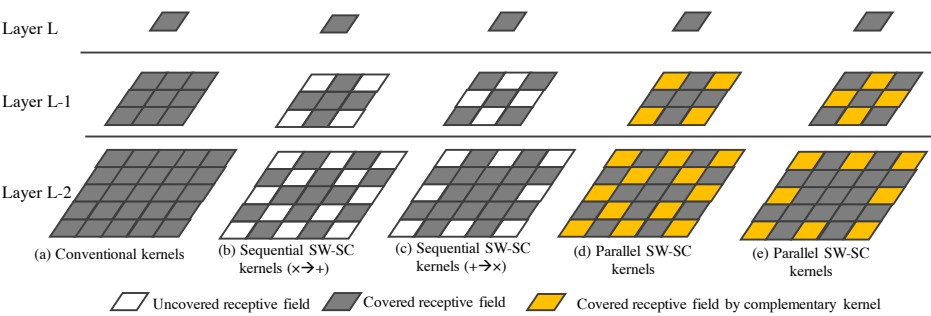

Figure 3: Receptive fields of different convolutional kernels of a location at a layer (across multiple channels). (a) Conventional kernels. (b) and (c) apply SW-SC kernels *sequentially* ((b) applies $\times$-shape at layer $L-2$ and $+$-shape at layer $L-1$, (c) reverses the order in (b)). (d) and (e) use SW-SC kernels. Due to the fact that feature maps from the previous layer are extracted by either even-indexed sparse kernels or odd-indexed sparse kernels, some uncovered receptive field can be recovered by complementary kernels (highlighted by yellow color).

Based on these observations, we favor increasing the number of kernels in a convolutional layer rather than to increase the depth of network. Increasing the number of kernels in a layer also potentially increase the capability of feature expressivity of the network due to the fact that each convolutional kernel can combine more input feature maps from the previous layers to represent more complex features[4]. Whereas, if we use deeper networks, each layer might be trapped into a bad feature representation due to the limitation of sparse kernels. The empirical experiments we performed show that, in most of the cases, both wider and deeper networks were able to achieve better classification accuracy than baseline, and the wider networks are the best in accuracy (see Section 4.1.3).

To increase the number of kernels in a layer, we use a control parameter $w$ for all layers in a network, i.e., the increasing ratio is homogeneous across the whole network. Take ResNet-18 for ImageNet as an example (He et al. (2016a)), the original number of parameters and FLOPs per pixel of a residual block is (a residual block contains two $3 \times 3$ convolutional layers in ResNet-18):

$$3 \times 3 \times C \times C \times 2, \tag{3}$$

where $C$ is the number of input feature maps and also the number of kernels in the convolutional layers for simplicity.

For our SW-SC convolutional layer, with the control parameter, $w$, the number of parameters and FLOPs per pixel would be:

$$5 \times (Cw) \times (Cw) \times 2, \tag{4}$$

where 5 comes from five active weights in a $3 \times 3$ kernel, and the number of input feature maps and the number of kernels in a convolutional layer become $Cw$ when we increase the number kernels in a layer. Thus, under the constraint of comparative resource budget, we set $w$ to $1.3125$ in this case.

## 4 EXPERIMENTAL RESULTS

To evaluate the proposed spatial-wise and channel-wise sparse-complementary (SW-SC and CW-SC) convolutional kernels, we applied them onto on state-of-the-art network architectures for the image classification task, including ResNet (He et al. (2016a;b)) and DenseNet (Huang et al. (2017)) for the CIFAR-10/100 (Krizhevsky & Hinton (2009)) and ImageNet-1K (Russakovsky et al. (2015)) datasets. For all experiments, we replace all $3 \times 3$ and $1 \times 1$ kernels by SW-SC and CW-SC convolutional kernels, respectively. We add a suffix $-sc$ to refer to our models with either SW-SC, CW-SC, or both SW-SC and CW-SC for both deeper and wider networks, and we apply the same increasing ratio ($w$) for number of kernels for all convolutional layers in a CNN. The setting of $w$ is varied for different networks because $w$ is derived to match the resource usage to baselines (as the example in Section 3.2). For ResNet, $w$ is applied for all convolutional layers in residual blocks and the first $7 \times 7$ convolution for the ImageNet dataset; for DenseNet, $w$ is directly applied on the growth rate, which is the increasing ratio of channels for each layer. All experiments are completed by Tensorpack (Wu (2017)), a high-level API for Tensorflow (Abadi et al. (2015)).

### 4.1 EVALUATION ON CIFAR-10/CIFAR-100

For the CIFAR-10/CIFAR-100 datasets, we evaluate three perspectives, including (I) classification accuracy, (II) accuracy and effective FLOPs of our deeper and wider networks and (III) the advantages of having complementary kernels in the same layers.

### 4.1.1 NETWORKS AND TRAINING DETAILS

We use pre-activation ResNet for CIFAR-10/100 (He et al. (2016b)), and we train all ResNets based on the setting of original work, i.e., data augmentation includes standard translation augmentation and random flipping in horizontal direction, image standardization is applied, and weights are initialized based on the method in He et al. (2015); however, we use whole 50k training images for training rather than splitting training data into 45k and 5k images as the original work did. We

---

[4]More kernels result in more output channels and hence more input feature maps are at next layer.

trained 185 epochs for ResNets with batch size 128, and the momentum is 0.9 and the weight decay is 0.0001; learning rate is set to 0.01 in the beginning for warm up and then changes to 0.1 after the first epoch; then, the learning rate is dropped 10 times at 91 and 137 epochs. For DenseNet, we adopt bottleneck and compression setting in DenseNet (Huang et al. (2017)) and the training setting of original paper. Most settings are identical to ResNet with following exceptions, including the trained epochs, batch size and learning rate. The trained epochs are 300 with batch size 64, and the initial learning rate is 0.1 without warming up and is reduced by 10 times at 150 and 225 epochs.

### 4.1.2 CLASSIFICATION ACCURACY ON CIFAR-10/100

Figure 4 compares baselines and our deeper and wider networks for all ResNets. We apply only SW-SC on ResNets since only $3 \times 3$ convolutional layer in the network. Our wider networks (ResNet-sc-wider) consistently surpass the baselines by $\sim 0.5\%$ and $\sim 1\%$ accuracy improvement on the CIFAR-10 and CIFAR-100 datasets across different depths of the ResNet networks, respectively, without introducing any complexity. Thus, under identical FLOPs and parameters, our models can utilize the models more efficient than the baselines, which indicates that having more simpler kernels can achieve better accuracy (See Table 8 in the supplementary section for details.). Furthermore, for both CIFAR-10 and CIFAR-100, our wider ResNet-110-sc ($w = 1.3125$) achieves better performance than ResNet-164, which increases $1.5\times$ more layers as compared to ResNet-110. Thus, it again justifies the proposed sparse-complementary kernels can achieve better model utilization. On the other hand, for DenseNet, we apply both SW-SC and CW-SC on $3 \times 3$ and $1 \times 1$ convolutional layers respectively and the results are shown in 2. For CIFAR-10, we see minor performance degradation but we observe large improvement on CIFAR-100 dataset ($0.74\%$); thus, our model is still better than the baseline in DenseNet architecture since CIFAR-100 is a more difficult dataset in comparison with CIFAR-10.

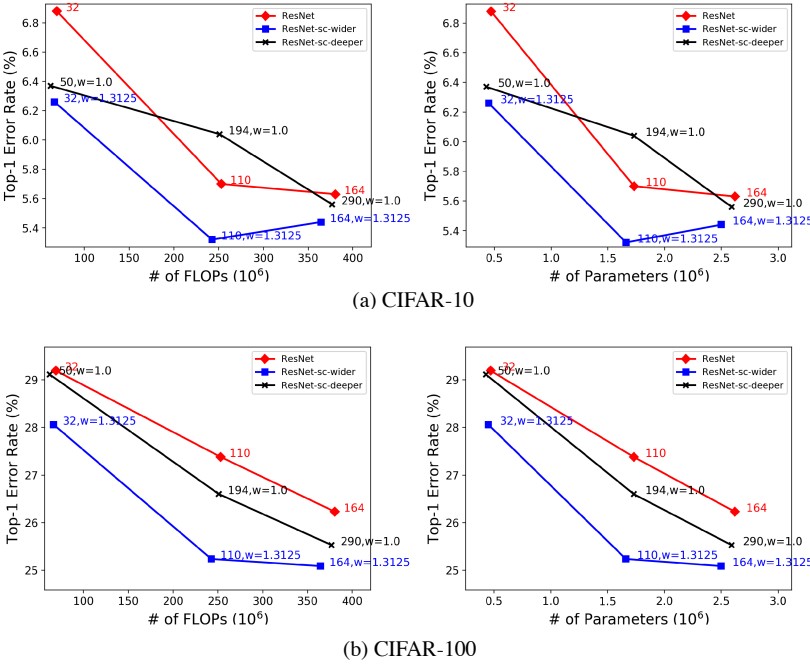

Figure 4: Error rate (%), FLOPs and # of parameters on the CIFAR-10/100 (C10/C100) datasets, including baselines and our wider and deeper networks.

### 4.1.3 COMPARISON BETWEEN OUR DEEPER AND WIDER NETWORKS

Figure 4 also compares our wider and deeper networks, and the wider networks are always better than the deeper networks. Thus, it justifies that by giving the same resource budget, the wider

Table 2: Error rate (%), FLOPs and # of parameters of DenseNet on the CIFAR-10/100 (C10/C100) and ImageNet datasets. **Bold** number indicates the best performance among the networks with similar complexity.

| Network | C10 | C100 | ImageNet | FLOPs ($\times 10^9$) | Params ($\times 10^6$) |
|---|---|---|---|---|---|
| DenseNet-121 ($k = 12$) | **4.01%** | 21.55% | − | 0.29 | 0.77 |
| DenseNet-121-sc ($k = 12, w = 1.375$) | 4.24% | **20.81%** | − | 0.30 | 0.78 |
| DenseNet-121 ($k = 32$) | − | − | 25.13% | 2.73 | 7.97 |
| DenseNet-121-sc ($k = 32, w = 1.3125$) | − | − | **24.85%** | 2.75 | 8.16 |

Table 3: Effective FLOPs ratio of wider and deeper networks (wider/deeper) at the same FLOPs. Effective FLOPs = $\frac{FLOPs}{GPU\ Running\ Time}$.

| Configuration | Effective FLOPs Ratio |
|---|---|
| ResNet-32-sc (w = 1.3125) / ResNet-50-sc (w = 1.0) | 1.085 |
| ResNet-110-sc (w = 1.3125) / ResNet-194-sc (w = 1.0) | 1.080 |
| ResNet-164-sc (w = 1.3125) / ResNet-290-sc (w = 1.0) | 1.079 |

The values larger than 1 mean wider networks achieve higher effective FLOPs.

network configuration can be more efficient than the deeper network. Since by having more convolutional kernels, we also extend the dimension of feature maps for exploring better feature representation at each layer rather than searching better feature representation at deeper layers.

On the other hand, as we discussed in section 3.2, for NVIDIA GPU, the wider networks achieved effective FLOPs than the deeper networks. Table 3 shows the ratio of effective FLOPs between wider and deeper SW-SC ResNet at the same FLOPs, and no matter the depth of base networks, the wider ones are always efficient than the deeper networks.

This is because current GPU is Single-Instruction-Multiple-Thread (SIMT) architecture, which is more suitable for computing the same instruction concurrently. That is, GPU can exploit higher parallelism when executing more identical instructions. Hence, if we have more kernels in the same convolutional layer, GPU can potentially dispatch all kernels onto GPU's cores to achieve better hardware utilization for faster speed.

### 4.1.4 ANALYSIS ON SPARSE-COMPLEMENTARY KERNEL

Table 4 shows the advantages in having complementary kernels at one convolutional layer rather than separating them into consecutive layers (Fan et al. (2017)). Sparse-complementary convolution helps the next layer mixing the information from both phases to provide better feature representation (larger receptive field as shown in Fig. 3). We found out that applying complementary kernels in the same layer can provide a better growth rate of receptive field, i.e., given the same number of layers, the receptive field with our approach can be larger. So, we achieve higher improvement on shallower networks, like ResNet-32 and ResNet-110; however, for ResNet-164, the deeper networks might compensate insufficient receptive field by its depth, both configurations achieve comparative results.

### 4.2 EVALUATION ON IMAGENET

We also apply our SW-SC and CW-SC kernels on ResNet and DenseNet for the ImageNet dataset, and our networks consistently achieve better accuracy under similar FLOPs and parameters.

### 4.2.1 NETWORKS AND TRAINING DETAILS

We use normal ResNet[5] (He et al. (2016a)) for the experiments on the ImageNet dataset since we found out that the normal ResNet achieves better accuracy than pre-activation ResNet on ImageNet,

---

[5]The normal ResNet means that the shortcut is added after batch normalization but before activation function.

Table 4: Error rate (%) among networks on the CIFAR-10/100 (C10/C100) datasets. **Bold** number indicates the best performance among the networks with similar complexity. $w$ of all models are set to 1.3125.

| Network | C10 | C100 |
|---|---|---|
| ResNet-32-sc | **6.26%** | **28.06%** |
| ResNet-32-sc-seq | 6.36% | 28.47% |
| ResNet-110-sc | **5.32%** | **25.24%** |
| ResNet-110-sc-seq | 5.51% | 25.77% |
| ResNet-164-sc | 5.44% | **25.09%** |
| ResNet-164-sc-seq | **5.31%** | 25.15% |

and we would like to evaluate performance improvement on top of higher accuracy model by our method. We train ResNet based on original work but with a minor modification: inside a residual block, the $\gamma$ parameter in the last batch normalization layer is initialized to zero (Goyal et al. (2017)), which lets the information flows fro identity shortcut in the beginning. We also apply color, scale and shift augmentation for training data (Wu (2017); Gross & Wilber (2016); Szegedy et al. (2015)). For both ResNet and DenseNet, we initialize weights based on the method in He et al. (2015) and train 110 epochs with total batch size 256. The momentum is 0.9 and the weight decay is 0.0001; initial learning rate starts is 0.1 and then it is dropped 10 times at 30, 60, 85, 95 and 105 epoch.

### 4.2.2 CLASSIFICATION ACCURACY ON IMAGENET

Figure 5 compares baselines and our models. To evaluate the trained model, we resize the shorter side of an image to 256 with the same aspect ratio, and then crop center region, $224 \times 224$, to evaluate top-1 error rate. For ResNet-18 and ResNet-34, we only apply SW-SC kernels since the residual block is composed of two $3 \times 3$ convolutional layers; for other ResNets and DenseNet, both SW-SC and CW-SC kernels are applied. Again, we achieve significant improvement on ResNet-18 and ResNet-34 when applying our proposed kernels with $w = 1.3125$: more than $1\%$ top-1 accuracy as compared to baselines. On the other hand, for ResNet-50, ResNet-101, ResNet-152 and DenseNet-121, those are high accuracy deep CNNs, it is difficult to improve their performance by simply modifying their micro-architectures. E.g., ResNet-152 outperforms ResNet-101 only $0.44\%$ top-1 accuracy, but ResNet-152 increase $1.5\times$ more layers, which is about $1.5\times$ FLOPs and parameters, to ResNet-101. Our models can consistently provide better top-1 accuracy, and our ResNet-101-sc improves $0.33\%$ top-1 accuracy, which boosts its performance close to ResNet-152 without introducing any overhead on FLOPs and parameters as compared to original ResNet-101; and our ResNet-152-sc provides another $\sim 0.2\%$ improvement (See Table 10 in the supplementary section for details .). Table 2 shows the same trend on DenseNet-121; without bringing any overhead, our model gains $0.3\%$ top-1 accuracy.

### 4.3 BENCHMARK OF SPARSE-COMPLEMENTARY CONVOLUTION ON NVIDIA GPU

In order to practice our SW-SC convolution on NVIDIA GPU with CUDNN, which is optimized for current state-of-the-art networks and convolutional kernels, we implemented our convolutions and archived competitive speed under the same computational load. All baselines' performance measurement relied on CUDNN 6.0. Thanks to the deterministic sparsity, we can easily revise original *im2col* routine to lower an input tensor into a matrix efficiently and then also exploit efficient matrix multiplication on GPU. We use NVIDIA K80 and Caffe framework (Jia et al. (2014)) for our speed benchmark. Table 5 compares the speedup ratio over the baselines under identical FLOPs; the sparse-complementary kernels achieve competitive speed against to highly-optimized CUDNN (Lavin & Gray (2016)).

### 4.4 DISCUSSION ON PERFORMANCE BOOST WITH SIMPLE KERNELS

In this chapter, we would like to address the possibility of performance increase with other types of simple kernels. Our proposal which increases accuracy based on the increased number of kernels

Table 5: GPU speed comparison of SW-SC and regular convolution under similiar computations.

| Configuration | Input tensor $N \times C \times H \times W^{*}$ | Convolutional layer $k \times k$, $K^{\dagger}$ | Speed up |
|---|---|---|---|
| A | $32 \times 672 \times 14 \times 14$ $32 \times 512 \times 14 \times 14$ | $3 \times 3$, 672 SW-SC $3 \times 3$, 512 | $1.00\times$ |
| B | $32 \times 672 \times 7 \times 7$ $32 \times 512 \times 7 \times 7$ | $3 \times 3$, 672 SW-SC $3 \times 3$, 512 | $0.99\times$ |
| C | $32 \times 1344 \times 7 \times 7$ $32 \times 1024 \times 7 \times 7$ | $3 \times 3$, 1344 SW-SC $3 \times 3$, 1024 | $\mathbf{1.21\times}$ |

$^{*}$: $N$: batch size, $C$: channels, $H$: height, $W$: width.

$^{\dagger}$: $k$: kernel size, $K$: number of kernels.

Stride and padding in convolutional layers are 1; $w$ is set to 1.3125.

relies on computation savings from its simpler convolution compared to $3 \times 3$ regular convolution. It means that $1 \times 3$, $3 \times 1$ kernels (Ioannou et al. (2016)) would have more chances to increase the number of kernels for boosting the accuracy consequently. In our additional experiments, as expected, these kernels showed the similar performance gain as well, since those kernels also have full receptive coverage of regular convolution like our complementary kernels. However, in this paper, we focused on the effects sparse-complementary kernel for some reasons[6], therefore, comparing benefits of different shapes of kernels is not part of purpose of this paper. But it seem clear that our performance boost proposal can be applicable for other types of simpler kernels, including $1 \times 3$, $3 \times 1$ and mixed-shape kernels, eventually.

## 4.5    COMPARISON WITH OTHER WORKS

To validate our kernel design, we compared our work with Ioannou et al. (2016), which uses $1 \times k$, $k \times 1$, and $1 \times 1$ convolutions to replace $3 \times 3$ convolutions. We use ResNet-50 as the case study (we trained the networks by ourselves), and the results are shown in Table 6. Under bottleneck residual block used in ResNet-50, the features are embedded into low-dimension space before applying non $1 \times 1$ convolutions; thus, $1 \times 3$ and $3 \times 1$ convolutions are not enough to approximate original $3 \times 3$ for better performance in low-dimension space. Even with wider network setting, the performance is worse than baseline. (Note that $1 \times 3$ and $3 \times 1$ are rank-1 kernels and they are not complemented.). On the other hand, our SW-SC kernels are rank-2 (+-shape) and rank-3 ($\times$-shape), which provide

---

[6]One of the reason is the complementary $\times$ and $+$ kernels are more suitable for computer vision task by nature. We are investigating its potential for different tasks such as localization, segmentation, etc. in parallel.

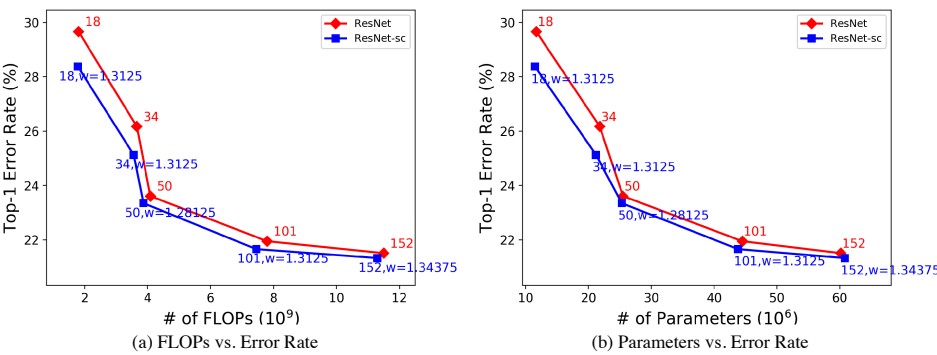

(a) FLOPs vs. Error Rate          (b) Parameters vs. Error Rate

Figure 5: Comparison between baselines and our models for all ResNets on ImageNet. Our models consistently provide better accuracy and model utilization.

better approximation; furthermore, our kernels are complementary, they are able to complement each other to learn better feature representation for higher accuracy.

On the other hand, interleaved group convolution (IGC) (Zhang et al. (2017)) uses group convolution with permutation to leverage the shortage of applying group convolution consecutively, and hence, they achieve better parameter utilization. Logically, CW-SC might be reduced to IGC whose group numbers are 2 for both group convolutions; however, there are a little different in implementation. IGC splits a group convolution and a permutation into two stages but CW-SC embeds the permutation into convolution, which provides more advantages in hardware implementation (ASIC and FPGA). Permutation process usually involves data movement and hence a temporary buffer might be required to swap data to different addresses. Thus, by embedding the permutation into convolution, our CW-SC has the potential to be more efficient than IGC in implementation even though they achieve identical algorithmic performance. We compare our SW-SC with IGC and the results are shown in Table 7. We find that the SW-SC surpasses IGC by 1 2% points except for the extreme case (IGC-L100M2), which is similar to XceptionNet. Furthermore, we also integrated IGC with our SW-SC (ResNet-18-sc-IGC-L16M16 ($w = 1.5$), we can achieve another 1.8% improvement over original IGC-L16M16+Identity, which justify the orthogonality between two works (If we compare with the reproduced IGC-L16M16, we achieve 2.4% improvement without any overhead on computations and parameters.).

Table 6: Error rate (%), FLOPs and # of parameters among networks on ImageNet. Single crop validation errors on a $224 \times 224$ crop is reported. **Bold** number indicates the best performance among the networks with similar complexity.

| Network | Top-1 | Top-5 | FLOPs ($\times 10^9$) | Params ($\times 10^6$) |
|---|---|---|---|---|
| ResNet-50 | 23.61% | 6.85% | 4.09 | 25.55 |
| Ioannou et al. (2016) ($w = 1.125$) | 24.46% | 7.42% | 3.86 | 24.10 |
| Ioannou et al. (2016)[*] ($w = 1.1875$) | 24.17% | 7.23% | 4.00 | 24.93 |
| ResNet-50-sc ($w = 1.28125$) (Ours) | **23.35%** | **6.74%** | 3.87 | 25.25 |

[*]:Only using $1 \times 3$ and $3 \times 1$ convolutions to replace $3 \times 3$ convolutions, so the width ratio is increased.

Table 7: Error rate (%), FLOPs and # of parameters among networks on ImageNet. Single crop validation errors on a $224 \times 224$ crop is reported.

| Network | Top-1 | Top-5 | FLOPs ($\times 10^9$) | Params ($\times 10^6$) |
|---|---|---|---|---|
| ResNet-18 | 29.66% | 10.50% | 1.81 | 11.69 |
| IGC-L4M32+Ident.(Zhang et al. (2017))[*] | 30.77% | 10.99% | 1.90 | 11.21 |
| IGC-L16M16+Ident.(Zhang et al. (2017))[*] | 29.40% | 10.99% | 2.20 | 11.33 |
| IGC-L16M16+Ident.(Zhang et al. (2017))[†] | 30.03% | 11.21% | 2.26 | 13.53 |
| IGC-L100M2+Ident.(Zhang et al. (2017))[*] | 26.95% | 8.92% | 1.3 | 8.61 |
| ResNet-18-sc ($w = 1.3125$) (Ours) | 28.06% | 9.80% | 1.79 | 11.51 |
| ResNet-18-sc-IGC-L16M16 ($w = 1.5$) | 27.59% | 9.48% | 2.35 | 13.15 |

[*]:Number reported in their paper. [†]: Reproduced by using Tensorpack.

## 5 CONCLUSION

We propose an efficient way to utilize model parameters for better accuracy without increasing any computational complexity and parameters. Due to the fact that the proposed sparse-complementary kernel has almost perfect coverage of receptive field only with half of computations, it achieves better results when we increase the number of kernels to match the computation amounts compared to regular convolution. Under the same computational budgets, we always achieve better performance in the classification task. We also demonstrate that the sparse-complementary convolutions can achieve competitive running speed on GPU in practice for most cases. Since our proposed performance boost using simpler kernel shows promising result with high model utilization, it will be worth investigating other types of simpler kernels for improving performance.

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

# 6 SUPPLEMENTARY MATERIALS

## 6.1 MORE ANALYSIS ON SW-SC AND CW-SC

Table 8: Error rate (%), FLOPs and # of parameters among networks on the CIFAR-10/100 (C10/C100) datasets. **Bold** number indicates the best performance among the networks with similar complexity.

| Network | C10 | C100 | FLOPs ($\times 10^6$) | Params ($\times 10^6$) |
|---|---|---|---|---|
| ResNet-32 | 6.88% | 29.19% | 69.1 | 0.47 |
| ResNet-50-sc ($w = 1.0$) | 6.37% | 29.11% | 62.3 | 0.43 |
| ResNet-32-sc-comb ($w = 1.875$) | 6.53% | 28.94% | 68.6 | 0.47 |
| ResNet-32-sc[†] ($w = 1.3125$) | 6.77% | 28.83% | 66.5 | 0.45 |
| ResNet-32-sc ($w = 1.0$) | 7.13% | 30.43% | 38.7 | 0.26 |
| ResNet-32-sc ($w = 1.3125$) | **6.26%** | **28.06%** | 66.5 | 0.45 |
| ResNet-110 | 5.70% | 27.38% | 253.1 | 1.73 |
| ResNet-194-sc ($w = 1.0$) | 6.04% | 26.60% | 251.1 | 1.73 |
| ResNet-110-sc-comb ($w = 1.875$) | 5.39% | 25.65% | 248.3 | 1.72 |
| ResNet-110-sc[†] ($w = 1.3125$) | 5.56% | 25.72% | 242.6 | 1.66 |
| ResNet-110-sc ($w = 1.0$) | 5.51% | 27.85% | 141.0 | 0.966 |
| ResNet-110-sc ($w = 1.3125$) | **5.32%** | **25.24%** | 242.6 | 1.66 |
| ResNet-164 | 5.63% | 26.23% | 380.6 | 2.62 |
| ResNet-290-sc ($w = 1.0$) | 5.56% | 25.53% | 376.9 | 2.59 |
| ResNet-164-sc-comb ($w = 1.875$) | 5.54% | 25.31% | 372.7 | 2.57 |
| ResNet-164-sc[†] ($w = 1.3125$) | 5.87% | 25.59% | 364.6 | 2.50 |
| ResNet-164-sc ($w = 1.0$) | 5.93% | 26.60% | 211.7 | 1.46 |
| ResNet-164-sc ($w = 1.3125$) | **5.44%** | **25.09%** | 364.6 | 2.50 |

-comb: combine SW-SC and CW-SC., [†]:Use the kernel shape in Fig. 6 for SW-SC.

**SW-SC kernel** The SW-Sc kernels are designed to approximate the receptive field of original dense kernels; however, by simply replacing dense kernels by SW-SC, we might degrade algorithmic performance but reduce model complexity. Table 8 compares the original ResNets and the ones use SW-SC without increasing the width ($w = 1.0$). In most case, SW-SC ($w = 1.0$) degrades accuracy slightly and sometimes it also improves the performance (ResNet-110 on CIFAR-10); however, SW-SC ($w = 1.0$) reduces the FLOPs and parameters about $1.8\times$, which increases model utilization. Table 9 shows real speedup of SW-SC ($w = 1.0$) over dense kernels when given the same configuration of an input tensor and a convolutional layer, and the used deep learning framework and measurement methods are identical to section 4.3. Our SW-SC convolution achieve $1.8\times$ speedup theoretically, and our implementation can achieve close speed up as compared to highly-optimized CUDNN when an input tensor with more channels (e.g. configuration D, E and F.).

**Combination of SW-SC and CW-SC** Due to the orthogonality of SW-SC and CW-SC, we combine them together and evaluate them on CIFAR-10/100. We use ResNet as the base network and use both SW-SC and CW-SC on every $3 \times 3$ convolutions in the residual block. Thanks to the saving on CW-SC, we can enlarge the network to $w = 1.875$ (sc-comb); however, each kernel will be too sparse to capture meaningful features (about 25% parameters of the original kernel); therefore, even with more kernels, its performance is only competitive to the one uses only SW-SC on $3 \times 3$ kernels (See Table 8 for details).

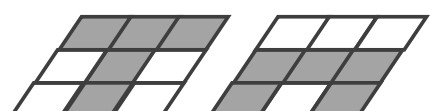

Figure 6: Another SW-SC kernels.

**Other shapes of SW-SC** In this paper, for $3 \times 3$ kernel, we propose to use $+$ and $\times$ shape as our base kernels; however, other shapes of sparse kernels might be able to achieve the same receptive field. Thus, we use another set of SW-SC kernels (SW-SC$^\dagger$) to validate our selection is the better choice according to the nature of computer vision application. SW-SC$^\dagger$ is composed of the shapes in Fig. 6 (both are rank-2). As shown in Table 8, it empirically justifies the inappropriate selection of sparse-complementary kernels degrades the performance.

Table 9: GPU speed up of SW-SC convolution to conventional convolution.

| Configuration | Input tensor $N \times C \times H \times W^*$ | Convolutional layer $k \times k,\ K^\dagger$ | Speedup |
|---|---|---|---|
| A | $32 \times 256 \times 28 \times 28$ | $3 \times 3,\ 256$ | $1.22\times$ |
| B | $32 \times 256 \times 28 \times 28$ | $3 \times 3,\ 512$ | $1.40\times$ |
| C | $32 \times 512 \times 14 \times 14$ | $3 \times 3,\ 256$ | $1.48\times$ |
| D | $32 \times 512 \times 14 \times 14$ | $3 \times 3,\ 512$ | $\mathbf{1.77\times}$ |
| E | $32 \times 1024 \times 7 \times 7$ | $3 \times 3,\ 512$ | $\mathbf{1.68\times}$ |
| F | $32 \times 1024 \times 7 \times 7$ | $3 \times 3,\ 1024$ | $\mathbf{1.98\times}$ |

$^*$: $N$: batch size, $C$: channels, $H$: height, $W$: width.

$^\dagger$: $k$: kernel size, $K$: number of kernels.

## 6.2 EXPERIMENTAL RESULTS ON IMAGENET

Table 10 shows the details on the experiments for ImageNet.

Table 10: Error rate (%), FLOPs and # of parameters among networks on ImageNet. Single crop validation errors on a $224 \times 224$ crop is reported. **Bold** number indicates the best performance among the networks with similar complexity.

| Network | Top-1 | Top-5 | FLOPs ($\times 10^9$) | Params ($\times 10^6$) |
|---|---|---|---|---|
| ResNet-18 | 29.66% | 10.50% | 1.81 | 11.69 |
| ResNet-18-sc ($w = 1.3125$) | **28.06%** | **9.80%** | 1.79 | 11.51 |
| ResNet-34 | 26.17% | 8.56% | 3.66 | 21.80 |
| ResNet-34-sc ($w = 1.3125$) | **25.13%** | **7.94%** | 3.56 | 21.18 |
| ResNet-50 | 23.61% | 6.85% | 4.09 | 25.55 |
| ResNet-50-sc ($w = 1.28125$) | **23.35%** | **6.74%** | 3.87 | 25.25 |
| ResNet-101 | 21.95% | 6.04% | 7.80 | 44.54 |
| ResNet-101-sc ($w = 1.3125$) | **21.62%** | **5.89%** | 7.45 | 43.77 |
| ResNet-152 | 21.51% | 5.78% | 11.51 | 60.19 |
| ResNet-152-sc ($w = 1.34375$) | **21.33%** | **5.64%** | 11.35 | 60.79 |

