# OpenReview forum: "Sparse-Complementary Convolution for Efficient Model Utilization on CNNs"
_ICLR.cc/2018/Conference — Reject_

### Official Review · AnonReviewer1 · 2017-11-27
**Very close to mixed-shaped kernels and interleaved group convolutions**

**Rating:** 6
**Confidence:** 5

**Review:**



This paper presented interesting ideas to reduce the redundancy in convolution kernels. They are very close to existing algorithms.

(1)	The SW-SC kernel (Figure 2 (a)) is an extension of the existing shaped kernel (Figure 1 (c)).
(2)	The CW-SC kernel (Figure 2 (c)) is very similar to interleaved group convolutions. The CW-SC kernel can be regarded as a redundant version of interleaved group convolutions [1].

I would like to see more discussions on the relation to these methods and more strong arguments for convincing reviewers to accept this paper.

[1] Interleaved Group Convolutions. Ting Zhang, Guo-Jun Qi, Bin Xiao, and Jingdong Wang. ICCV 2017. http://openaccess.thecvf.com/content_ICCV_2017/papers/Zhang_Interleaved_Group_Convolutions_ICCV_2017_paper.pdf

---

> ### Author Response · Authors · 2017-12-23
> **Response**
>
> Thanks for your comments.
>
> Interleaved group convolution (IGC) is composed of two group convolutions followed by permutations, and the group numbers of the first group convolution and the second group convolution are correlated. (The group numbers of the second group convolution is the factor of the number of channels in a group of the first group convolution.)
>
> Therefore, one clear difference of our paper from IGC is SW-SC kernels (mixed-shape kernels), which are orthogonal to each other; on the other hand, our CW-SC is a group convolution embeds the permutation on output feature maps; hence, CW-SC assures each feature map layer L+2 can get information from all feature maps at layer L.
>
> Logically, CW-SC might be reduced to IGC whose group numbers are 2 for both group convolutions; however, there are a little different in implementation. IGC splits a group convolution and a permutation into two stages but CW-SC embeds the permutation into convolution, which provides more advantages in hardware implementation (ASIC and FPGA). Permutation process usually involves data movement and hence a temporary buffer might be required to swap data to different addresses. Thus, by embedding the permutation into convolution, our CW-SC has the potential to be more efficient than IGC in implementation even though they achieve identical algorithmic performance.
>
> On the other hand, we compare our SW-SC with IGC under similar FLOPs and parameters for ResNet-18 network, our ResNet-18-sc (w=1.3125) outperforms IGCs (IGC-L4M32+Identity and IGC-L16M16+Identity) on the ImageNet dataset by ~1 to 2% without introducing extra FLOPs and parameters (see Table 7 in the revision for details) except for the extreme case of IGC (IGC-L100M2), which is similar to XceptionNet. Furthermore, we also integrate our SW-SC with IGC (IGC-L16M16+Identity), this combination boosts the performance by another 1.8% over original IGC, which justify the orthogonality between two works.

---

### Official Review · AnonReviewer2 · 2017-11-27
**An interesting idea, but generalization is not clear and the experiments not entirely convincing**

**Rating:** 5
**Confidence:** 4

**Review:**

This paper introduces a new design of kernels in convolutional neural networks. The idea is to have sparse but complementary kernels with predefined patterns, which altogether cover the same receptive field as dense kernels. Because of the sparsity of such kernels, deeper or wider networks can be designed at the same computational cost as networks with dense kernels.

Strengths:
- The complementary kernels come at no loss compare to standard ones
- The resulting wider networks can achieve better accuracies than the original ones

Weaknesses:
- The proposed patterns are clear for 3x3 kernels, but no solution is proposed for other dimensions
- The improvement over the baseline is not very impressive
- There is no comparison against other strategies, such as 1xk and kx1 kernels (e.g., Ioannou et al. 2016)

Detailed comments:
- The separation into + and x patterns is quite clear for 3x3 kernels. However, two such patterns would not be sufficient for 5x5 or 7x7 kernels. This idea would have more impact if it generalized to arbitrary kernel dimensions.

- The improvement over the original models are of the order of less than 1 percent. I understand that such improvements are not easy to achieve, but one could wonder if they are not due to the randomness of initialization/mini-batches. It would be more meaningful to report average accuracies and standard deviations over several runs of each experiment.

- Section 4.4 briefly discusses the comparison with using 3x1 and 1x3 kernels, mentioning that an empirical comparison is beyond the scope of this paper. To me, this comparison is a must. In fact, the discussion in this section is not very clear to me, as it mentions additional experiments that I could not find (maybe I misunderstood the authors). What I would like to see is the results of a model based on the method of Ioannou et al, 2016 with the same number of FLOPS.

- In Section 2, the authors review ideas of so-called random kernel sparsity. Note that the work of Wen et al., 2016, and that of Alvarez & Salzmann, NIPS 2016, do not really impose random sparsity, but rather aim to cancel out entire kernels, thus reducing the size of the model and not requiring implementation overhead. They also do not require pre-training and re-training, but just a single training procedure. Note also that these methods often tend not to decrease accuracy, but rather even increase it (by a similar magnitude to that in this paper), for a more compact model.

- In the context of random sparsity, it would be worth citing the work of Collins & Kohli, 2014, Memory Bounded Deep Convolutional Networks.

- I am not entirely convinced by the discussion of the grouped sparsity method in Section 3.1. In fact, the order of the channels is arbitrary, since the kernels are learnt. Therefore, it seems to me that they could achieve the same result. Maybe the authors can clarify this?

- Is there a particular reason why the central points appears in both complementary kernels (+ and x)?

- Why did the authors change the training procedure of ResNets slightly compared to the original paper, i.e., 50k training images instead of 45k training + 5k validation? Did the baseline (original model) reported here also use 50k? What would the results be with 45k?

- Fig. 5 is not entirely clear to me. What was the width of each layer? The original one or the modified one?

- It would be interesting to report the accuracy of a standard ResNet with 1.325*width as a comparison, as well as the runtime of such a model.

- In Table 4, I find it surprising that there is an actual speedup for the model with larger width. I would have expected the same runtime. How do the authors explain this?

---

> ### Author Response · Authors · 2017-12-23
> **Response**
>
> Thanks for reviewer's comments.
>
> This paper does not aim to achieve the significant improvement for the state-of-the-art CNN models but rather rethinks how do we design convolutional kernels. Do we always need dense kernels or we can use more sparse kernels to achieve better accuracy? The proposed method is orthogonal to any type of macro-architecture network design, (We select two state-of-the-art networks (ResNet and DenseNet) as our case study.)
>
> Our method is not restricted to 3x3 kernels. Based on the expressions in the manuscript, we define our sparse kernel by even- and odd-indexed, and then 3x3 will be a special case since its kernel shapes become `+` and `x`; therefore it can be extended to 5x5 or large kernels. Nonetheless, as suggested in the VGG16 paper, the 5x5 kernels can be achieved by two 3x3 kernels, and most of the state-of-the-art network only use 3x3 kernels. We also deploy it on AlexNet (which has 5x5 kernel) to validate our work can be extended to other kernel sizes.
>
> We compare our results with Ioannou et.al. 2016 (ICLR 2016) and interleaved group convolution (ICCV 2017) in Table 6 and 7 in the revision.
>
> Here are the response of detailed comments:
>
> 1. We deploy our idea on AlexNet, we replace all 3x3 and 5x5 kernels with our SW-SC kernels. Our AlexNet-sc-1.3125 achieves 1.5% accuracy improvement at the same computational costs, which justifies our approach can be deployed on larger kernel; however, 3x3 is the preferred.
>
> 2. Due to limited computing resource we had, we rerun the experiments of ResNet-101-sc-1.3125 two times, the average top-1 accuracy is 21.71%, which is still better than baseline 0.25%, by considering the performance difference of ResNet-101 and ResNet-152 is only 0.44%, which increases ~1.5x FLOPs and parameters to achieve so. Our approach still achieves gain with the same macro-network structure without increasing overhead on FLOPs and parameters.
>
> 3. We compare the work with Ioannou et al. 2016 on ResNet-50 (We trained models by ourselves). We evaluate two configurations of their work, (I) replace 3x3 convolutions by 1x3, 3x1 and 1x1 convolutions and (II) replace 3x3 convolution by only 1x3 and 3x1 convolutions. We also increase the width to align the FLOPs and parameters. Under the same FLOPs, their work achieved 24.46% and 24.17% top-1 error rate for configuration (I) and (II) respectively. The results are worse than the baseline and ours. The bottleneck topology in a residual block embeds the feature into low-dimension space, by simply using 1x3 and 3x1 convolutions is not sufficient to extract discriminative features in the low-dimension space (note that: 1x3 and 3x1 are rank-1 kernels and not complemented); however, our kernels are rank-2 and rank-3 kernels and they are complementary which provide better approximation; thus, our approach learns better feature representation for better performance. (See Table 6 in the revision.)
>
> 4. Yes, for those reference works (Wen et al and Alvarez & Salzmann), their sparsity is structural but still random, which means the sparsity are not fixed across different models. They show the speedup on CPU and GPU in their paper since CPU and GPU are flexible to prepare the data for different structural sparse convolution but for customized neural network accelerator, those random but structural sparse convolution will still bring the overhead in configuring the processing elements. Thus, we want our kernel to be deterministic across different models, and then we are able to provide the regularity for both GPU implementation and customized neural network accelerator, which might be realized by ASIC or FPGA; furthermore, more and more IoT devices are able to run inference on the edge; without deterministic sparsity, those edge devices might gain limited advantages from random but structural sparsity.
>
> 5. This paper is cited in the revision and discussed in 'Related Works' section.

---

> ### Author Response · Authors · 2017-12-23
> **Response (cont.)**
>
> 6. The major difference between CW-SC and group convolution is the order of output feature maps. CW-SC embeds a permutation into convolution. Thus, if two CW-SC layers are deployed consecutively, it is different from deploying two consecutive group convolutions. E.g., for CW-SC, every feature map at layer L+2 gets the information of all feature maps at layer L; however, for group convolution, the feature map at layer L+2 only gets the information from the first half or the second half feature maps at layer L since the group convolution always use the particular partition of feature maps. (The paper is also revised accordingly.)
>
> 7. The pattern design is inspired from two perspectives: (I) the observation from trained network with dense kernels, like VGG16 and VGG11, and (II) some preliminary empirical studies that the kernel without center point degrades performance.
>
> 8. We train all baselines and our methods by ourselves, and all experiments on CIFAR-10/100 use 50k images. Thus, it should be a fair comparison and we do not expect significant difference if we use 45k image to train and test the models.
>
> 9. We use Table 3 to replace original Figure 5 to clarify the results. We would like to compare the computational effectiveness between wider and deeper ResNets and both with SW-SC layers. For the wider networks, the w are set to 1.3125, for deeper networks, w are 1.0 but with more layers, and we tested on three configurations of base networks (ResNet-32, ResNet-110 and ResNet-164). The results show that no matter which configuration is, the wider network is always effective than deeper network at the same FLOPs. Please see Table 3 for details.
>
> 10. We can expect that ResNet (w=1.3125) outperforms our sparse models with the same number of kernels but ResNet (w=1.3125) costs more parameters and computations. It is a similar comparison between ResNet (w=1.0) and ResNet-sc (w=1.0), and we add this comparison in Table 8. The results show that ResNet-sc (w=1.0) reduces the FLOPs and parameters but slightly degrades the performance; furthermore, Table 9 shows the speed comparison. For most cases, our SW-SC achieves theoretical speedup (1.8x) or even better (please refer to bullet 11 for the details of benchmark.)
>
> 11. We would like to clarify how do we benchmark first. For the baseline, we directly use CUDNN 6.0 to run the CNN model without any modification; however, for our SW-SC approach, we customized our implementation with batched convolution to fully utilize GPU resource. Thus, maybe CUDNN is not fully optimized but we do not have the control to tune its performance since it is proprietary product from NVIDIA. Our goal is to provide a proof of concept that this type of deterministic is really helpful for GPU as compared to the kernels with random sparsity.

---

### Official Review · AnonReviewer3 · 2017-11-28
**This paper proposed a sparse-complementary convolution as an alternative to the convolution operation in deep networks. The paper is easy to follow and the idea is interesting. However, the novelty of the paper is limited and the experiments are not sufficient.**

**Rating:** 5
**Confidence:** 4

**Review:**

Summary:
This paper proposed a sparse-complementary convolution as an alternative to the convolution operation in deep networks. In this method, two new types of kernels are developed, namely the spatial-wise and channel-wise sparse-complementary kernels. The authors argue that the proposed kernels are able to cover the same receptive field as the regular convolution with almost half the parameters. By adding more filters or layers in the model while keeping the same FLOPs and parameters, the models with the proposed method outperform the regular convolution models. The paper is easy to follow and the idea is interesting. However, the novelty of the paper is limited and the experiments are not sufficient.

Strengths:
1. The authors proposed the sparse-complementary convolution to cover the same receptive field as the regular convolution.

2. The authors implement the proposed sparse-complementary convolution on NVIDIA GPU and achieved competitive speed under the same computational load to regular convolution.

3. The authors demonstrated that, given the same resource budget, the wider networks with the proposed method are more efficient than the deeper networks due to the nature of GPU parallel mechanism.

Weak points:

1. The novelty of this paper is limited. The main idea is to design complementary kernels that cover the same receptive field as the regular convolution. However, the performance improvement is marginal and may come from the benefit of wide networks rather than the proposed complementary kernels. Moreover, the experiments are not sufficient to support the arguments. For example, how is the performance of a model containing SW-SC or CW-SC without deepening or widening the networks? Without such experiment, it is unclear whether the improved performance comes from the sparse-complementary kernels or the increased number of kernels.

2. The relationship between the proposed spatial-wise kernels and the channel-wise kernels is not very clear. Which kernel is better and how to choose between them in a deep network? There is no experimental proof in the paper.

3. The proposed two kernels introduce sparsity in the spatial and channel dimension, respectively. The two methods are used separately. Is it possible to combine them together?

4. The proposed method only considers the “+-shape” and “x-shape” sparse pattern. Given the same receptive field with multiple complementary kernels, is the kernel shape important for the training? There is no experimental result to verify this.

5. As mentioned in the paper, there are many methods which introduce sparsity in the convolution layer, such as “random kernels”, “low-rank approximated kernels” and “mixed-shape kernels”. However, there is no experimental comparison with these methods.

6. In the paper, the author mentioned another sparse-complementary baseline (sc-seq), which applies sparse kernels sequentially. It yields smaller receptive field than the proposed method when the model depth is very small. Indeed, when the model goes deeper, the receptive field becomes very close to that of the proposed method. In the experiments, it is strange that this method can also achieve comparable or better results. So, what is the advantage of the proposed “sc” method compared to the “sc-seq” method?


8. Figure 5 is hard to understand. This figure only shows that training shallower networks is more effective than training the deeper networks on GPU. However, it does not mean training the wider networks is more efficient than training the deeper ones.

---

> ### Author Response · Authors · 2017-12-23
> **Response**
>
> Regarding the weak points addressed by the reviewer:
>
> We would like to restate our goals in this paper first. This paper does not aim to achieve significant improvement for the state-of-the-art CNN models but rethinking how do we design convolutional kernels. Do we always need dense kernels or we can use more sparse kernels to achieve better accuracy? The proposed method is orthogonal to any type of macro-architecture network design, (We select two state-of-the-art networks (ResNet and DenseNet) as our case study.) Furthermore, we add more experiments to study the proposed method and compare with recent work in mixed-shape kernel (ICLR2016) and interleaved group convolution (ICCV2017).
>
> Here are our response:
>
> 1. The goal of this paper aims to explore a better way to utilize the parameters in a CNN model; conventionally, dense kernels are used but there are too many redundancies (the redundant parameters still improve performance but it is marginal.) By simply using SW-SC kernels on the original networks, the performance would be degraded but complexity is also reduced; i.e., a trade-off between complexity and accuracy. We add the comparison between ResNet and ResNet-sc (w=1.0) to discuss the effect from SW-SC kernels (see Table 8 in supplementary section). As expected, ResNet-sc (w=1.0) reduces FLOPs and parameters with slight performance degradation.
>
> 2. SW-SC and CW-SC are orthogonal to each other, SW-SC sparsifies kernels in spatial-wise domain; on the other hand, the CW-SC sparsifies kernels in channel-wise domain. For our best practice, SW-SC is for kxk kernels (k > 1) and CW-SC for 1x1 kernels as we used in our all experiments. The reason is that for computer vision tasks, SW-SC is a simplified version to extract spatial context and CW-SC is a simplified version to fuse different feature maps. We also provide some results by combining them together and the results show that our original setting is simple and achieve the similar results (see Table 8 in the supplementary section.)
>
> 3. It is possible to combine with them since SW-SC and CW-SC are orthogonal. There is no conflict between them. However, by combining them together, the kernel might be too sparse to capture features. For the best practice, we deploy SW-SC for kxk kernels (k > 1) and CW-SC for 1x1 kernels. There are some preliminary experiments for combining them together at Table 8 in the supplementary section.
>
> 4. Ideally, if the sparse shapes are complementary, we can stack more layers to achieve the same receptive fields; however, the shape of base kernel still matters, we experiment another sparse and complementary kernels and the performance are degraded and it empirically justify that the shape of sparse kernel should follow the nature of computer vision to extract meaningful features. (See Table 8 and Figure 6 in the supplementary section.)
>
> 5. We compare the work with Ioannou et al. 2016 on ResNet-50 (We trained models by ourselves). We evaluate two configurations of their work, (I) replace 3x3 convolutions by 1x3, 3x1 and 1x1 convolutions and (II) replace 3x3 convolution by only 1x3 and 3x1 convolutions. We also increase the width to align the FLOPs and parameters. Under the same FLOPs, their work achieved 24.46% and 24.17% top-1 error rate for configuration (I) and (II) respectively. The results are worse than the baseline and ours. The bottleneck topology in a residual block embeds the feature into low-dimension space, by simply using 1x3 and 3x1 convolutions is not sufficient to extract discriminative features in the low-dimension space (note that: 1x3 and 3x1 are rank-1 kernels and not complemented); however, our kernels are rank-2 and rank-3 kernels and they are complementary which provide better approximation; thus, our approach learns better feature representation for better performance. (See Table 6 in the revision.)
>
> 6. As the reviewer pointed out, our approach performs better for shallow networks. `sc-seq` might achieve competitive performance when the network is deep. A shallower network use fewer resource in both training and deployment. E.g., the system required fewer buffers for storing the features in the intermediate layers and the number of data loading of weights is less as well. We also perform different widening ratio (2.625 and 3.9375) for ResNet-32-sc and ResNet-32-sc-seq on CIFAR-10/100, ResNet-32-sc always outperforms ResNet-32-sc-seq.
>
>
> 8. We use Table 3 to replace original Figure 5 to clarify the results. We would like to compare the computational effectiveness of wider and deeper ResNets and both with SW-SC layers. For the wider networks, the "w" are set to 1.3125, for deeper networks, "w" are 1.0 but with more layers, and we tested on three configurations of base networks (ResNet-32, ResNet-110 and ResNet-164). The results show that no matter which configuration is, the wider network is always effective than deeper network at the same FLOPs. Please see Table 3 for details.

---

### Author Response · Authors · 2017-12-23
**Revision is available and modifications are highlighted in blue color**

Thanks for all reviewers' feedback. The revision is available and modifications are highlighted in blue color

---

### Decision · Program_Chairs · 2018-01-29
**ICLR 2018 Conference Acceptance Decision**

**Decision:**

Reject

**Comment:**

The paper studies factorizations of convolutional kernels. The proposed kernels lead to theoretical and practical efficiency improvements, but these improvements are very, very limited (for instance, Figure 5). It remains unclear how they compare to popular alternative approaches such as group convolutions (used in ResNeXt) or depth-separable convolutions (used in MobileNet). The reviewers identify a variety of smaller issues with the manuscript.